# Safety and resource utilisation efficiency of semi-skeletonised versus skeletonised left internal mammary artery harvesting techniques: The BANGABANDHU study

Redoy Ranjan[1,2,3*], Aziz Momin[2], Riyaz A. Kaba[4], Gie Ken-Dror[1], Sanjay Kumar Raha[5], Md Kamrul Hasan[5], Venkatachalam Chandrasekaran[2], Asit Baran Adhikary[3,6]

**1** Department of Biological Sciences, Royal Holloway University of London, London, United Kingdom, **2** Department of Cardiac Surgery, St George's University Hospitals NHS Foundation Trust, London, United Kingdom, **3** Department of Cardiac Surgery, Bangabandhu Sheikh Mujib Medical University, Dhaka, Bangladesh, **4** Department of Cardiology, St George's University Hospitals NHS Foundation Trust, London, United Kingdom, **5** Department of Cardiac Surgery, National Institute of Cardiovascular Diseases, Dhaka, Bangladesh, **6** Department of Cardiac Surgery, Impulse Hospital & Research Centre, Dhaka, Bangladesh

* redoy_ranjan@bsmmu.edu.bd

## Abstract

### Background

The ideal harvesting techniques of the left internal mammary artery (LIMA) for coronary artery bypass graft (CABG) are elusive. We assessed the safety and resource utilisation efficiency of semi-skeletonised LIMA harvesting techniques, focusing on length, harvesting time, and the number of Ligaclips used compared to skeletonised techniques within a single surgeon's practice.

### Methods

The BANGABANDHU (Bangladeshi Atherosclerosis Biobank AND Hub) study was an ambispective observational cohort that evaluated age- and sex-matched 2209 adult Bangladeshi isolated CABG population from 1st January 2015 to 31 January 2025. Univariate analysis observed the difference pattern in the dataset, while multivariate logistic regression (LR) analysis identified the independent variables associated with the advantage of semi-skelitonised LIMA. The area under the receiver operating characteristic (AUROC) curve demonstrated the goodness-of-fit of the prediction model.

### Results

We evaluated 2209 age- and sex-matched adult isolated CABG patients (skeletonised LIMA; n = 1050 and semi-skeletonised LIMA; n = 1159) with identical comorbidities (EuroSCORE II, hypertension, diabetes, renal impairment, COPD, left main

**Data availability statement:** The data are not publicly available due to privacy or ethical restrictions; however, the data supporting current study findings are available on reasonable request from Dr Sanjoy Kumar Saha, Consultant of Cardiac Anaesthesia (email: sanjoydr@bsmmu.edu.bd), who holds the data and responds to external requests for data access. Further, we will ensure long-term data storage using institutional archives, reliable repositories, and cloud storage, as well as employ redundancy like multiple backups to enhance data availability.

**Funding:** This study was funded by a research grant from the Bangladesh Medical Research Council (BMRC), reference number BMRC/Research Grant/2025/191 (1-10). The funders had no role in the study design, data collection and analysis, the decision to publish, or the preparation of the manuscript.

**Competing interests:** The authors have declared that no competing interests exist.

and multivessel coronary artery disease) between study groups ($p > 0.05$). LIMA harvest time ($35.9 \pm 5.5$ vs $16.6 \pm 3.9$; $p < 0.001$) and number of used Ligaclip ($23.6 \pm 4.8$ vs $11.7 \pm 3.6$; $p < 0.001$) were significantly higher in the skeletonised compared to the semi-skeletonised LIMA sample. Furthermore, an age and sex-adjusted multivariate logistic regression model found LIMA harvest time (odds ratio [OR] 0.067, 95% CI 0.01–0.39; $p = 0.003$) and number of used Ligaclip (OR 0.561, 95% CI 0.41–0.76; $p < 0.001$) significantly lower among semi-skeletonised LIMA techniques.

## Conclusion

The semi-skeletonised LIMA technique is advantageous as it significantly reduces harvesting time and requires fewer Ligaclips compared to the skeletonised technique.

---

## Introduction

Coronary artery bypass graft (CABG) surgery frequently utilises the left internal mammary artery (LIMA) for revascularisation of the left anterior descending (LAD) artery in ischaemic heart disease (IHD) [1,2]. A skeletonised LIMA harvesting technique involves dissecting the mammary artery free from the surrounding fascia, veins, and surrounding adipose tissue of the chest wall, leaving only the LIMA itself, which is commonly utilised conduits in CABG surgery [3,4]. Previously published papers observed that the skeletonised LIMA conduits have a longer conduit length and better postoperative clinical and angiographic profile compared to the pedicled LIMA graft [3,5–7]. Further, several authors reported that it is associated with reduced sternal wound complications and mediastinitis, making it the preferred harvesting technique for patients undergoing CABG surgery, especially with bilateral internal mammary artery grafting [6–8]. However, skeletonised LIMA has potential risks, especially a higher conduit haematoma due to iatrogenic injury and a higher graft occlusion rate requiring repeat revascularisation. On top of that, the skeletonised LIMA harvest technique is time-consuming and requires more Ligaclip to occlude the branches, which costs more money [7–10].

Nowadays, cardiac surgeons have utilised a modified harvesting technique known as the semi-skeletonised technique, which involves harvesting the LIMA with the veins and thin rim of connective tissues that act as a protective shield from iatrogenic and inadvertent trauma, especially from electrocautery [11,12]. This semi-skeletonised harvesting technique was found to have minimal postoperative bleeding, and the total amount of bleeding was comparable to that of the skeletonised and pedicle techniques, regardless of pleural integrity [13]. Nevertheless, existing literature found less incidence of sternal wound infection in semi-skeletonised LIMA compared to the pedicled technique, which paradoxically suggests similar findings in the skeletonised LIMA procedure [12–15]. However, it is essential to note that the decision between skeletonised and semi-skeletonised mammary harvest depends on the surgeon's comfort level and expertise, and both techniques can be used effectively [11,14,15].

To the best of our knowledge, this is the first large-scale study comparing the skeletonised and semi-skeletonised LIMA harvesting techniques to evaluate the benefits of the semi-skeletonised LIMA harvesting techniques over skeletonised LIMA, with a secondary aim to identify early complications associated with the semi-skeletonised LIMA technique.

## Materials and methods

The BANGABANDHU (Bangladeshi Atherosclerosis Biobank AND Hub) is an ambispective observational study with protocols published elsewhere [16]. We evaluated 2209 age- and sex-matched adult IHD patients undergoing either primary or redo isolated coronary artery bypass graft (CABG) surgery in a single surgeon's practice from January 1, 2015, to January 31, 2025. We recruited the study population into two groups based on the stochastically performed LIMA harvest techniques: either the skeletonised or the semi-skeletonised group. We utilised a probabilistic sampling method. Prior to the recruitment of the study population, ethical clearance was obtained from the institutional review board of Bangabandhu Sheikh Mujib Medical University (BSMMU/2024/5390; date 21 May 2024), Bangladesh, and this study complied with the Declaration of Helsinki. Due to its ambispective nature, we recruited participants retrospectively from preexisting hospital databases and prospectively from 1 June 2024 until 31 January 2025 for research purposes. For retrospectively recruited participants, all data were fully anonymised before accessing the data, and informed consent was obtained during their follow-up visits, while prospective participants provided informed consent at the recruitment. To ensure quality control and data completeness, we utilised standardised data collection tools, which have been described elsewhere [16] and provided thorough training for data collectors who were post-graduate surgical trainees. Further, two independent investigators cross-verified the data integrity and applied statistical methods to assess completeness, reducing biases and enhancing reliability. Data were fully anonymised and encrypted to maintain confidentiality. We recruited extensive sociodemographic data, such as age, gender, comorbidities and details of LIMA harvesting technique, harvesting time, LIMA length, and number of Ligaclip used. Furthermore, we recorded complications related to the harvesting techniques, such as LIMA spasm, haematoma, and any chest re-exploration required due to bleeding. LIMA haematoma is defined as a localised intramural or extramural collection of blood within or around the adjacent soft tissue, appearing as a bulging, discoloured arterial segment that may compromise graft patency. Although no standard distance threshold defines a LIMA haematoma, we discarded the conduit if the haematoma extended beyond 2 cm and was likely to impair distal run-off, opting instead for alternative conduits such as the radial artery or saphenous vein. In this study, we ensured at least 1–2 cm of healthy distal LIMA with good run-off, as assessed visually, for safe anastomosis. Furthermore, LIMA spasm was defined as arterial blanching or narrowing resulting from a transient vasoreactive response to mechanical manipulation, excessive dissection, or thermal injury from cautery, which potentially reduces distal blood flow, either temporarily or persistently, thereby compromising conduit patency. In this study, we applied warm topical vasodilators, including papaverine, nitroglycerin, and calcium channel blockers (e.g., diltiazem), and ensured continuous hydration of the LIMA during harvesting. Additionally, intraluminal vasodilators, such as papaverine and nitroglycerin, were administered if needed to improve the spasm.

### LIMA harvesting technique

The LIMA harvest was conducted via a standard median sternotomy, beginning with the dissection of the parietal pleura, connective tissue, and endothoracic fascia using low-power electrocautery. The starting point of the LIMA harvest typically depends on the individual case, usually initiated at the 3rd or 4th intercostal space or the most visibly accessible part of the LIMA. For skeletonised LIMA, dissection proceeded meticulously along the adventitial layer, leading to the exposed entire artery down to its distal bifurcation and only the artery was harvested (Fig 1a,b). Side branches were carefully clipped, and the chest wall side of these branches was cauterised and clipped for optimal haemostasis, depending on the size of the branches. The semiskeletonised LIMA was excised as a veno-arterial pedicle with surrounding thin adipose tissue, keeping ~1 cm of tissue on either side of the LIMA, except for muscular support and endothoracic fascia (Fig 1c,d,e),

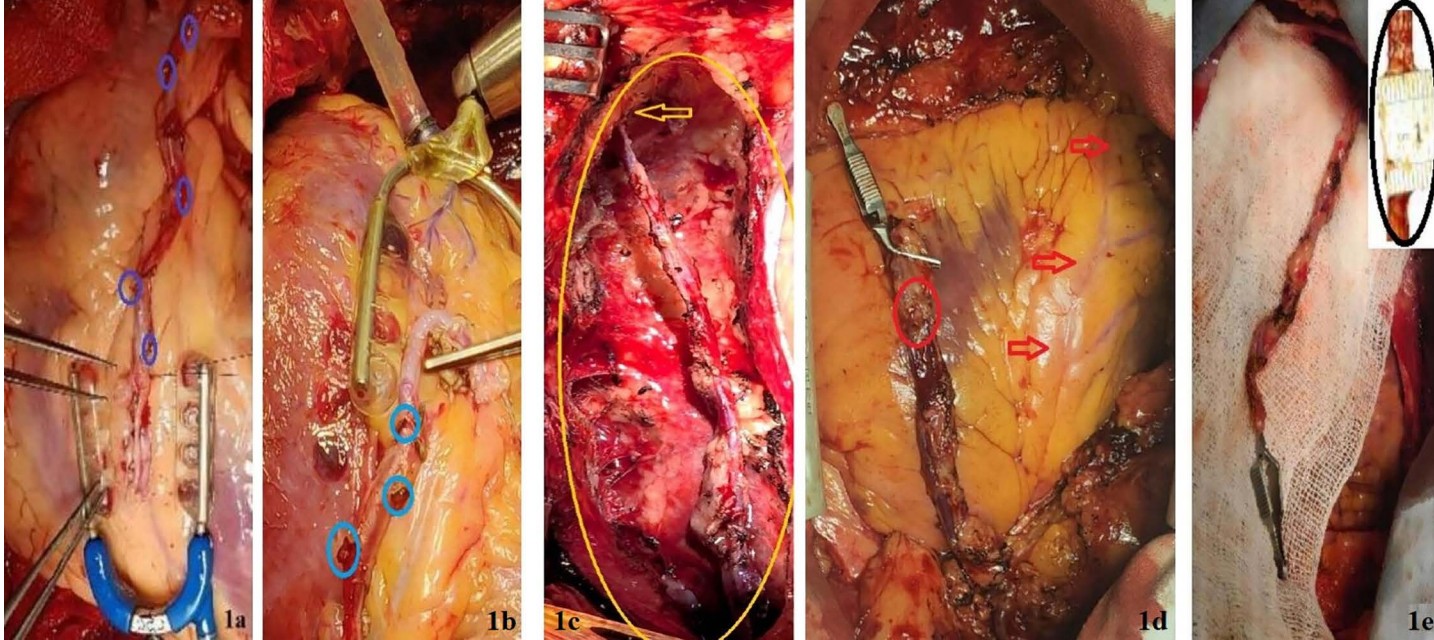

**Fig 1. LIMA harvesting techniques- skeletonised (1a, 1b) and semi-skeletonised (1c, 1d, 1e); Purple (1a) and Blue (1b) circles indicate the used Ligaclips.** 1c: The yellow arrow shows the distal end of LIMA just before the bifurcation; 1d: The red arrow shows the left anterior descending artery on a Dextrocardia, and the red circle shows the diameter of semi-skeletonised LIMA; 1e: The black circle indicates the diameter (1 cm) of semi-skeletonised LIMA.

as Horii and Suma described previously [17]. The initial incision of the pleura and endothoracic fascia was performed along the medial aspect of the accompanying vein, without an incision on the lateral side. The semi-skeletonised LIMA was then mobilised by scraping it from the endothoracic fascia with the cautery tip, while keeping it attached to the chest wall. For branches >1mm size, Ligaclips were applied on the LIMA side, while the thoracic side was electrocauterised. For intercostal branches <1 mm in size, coagulation was performed for approximately 3–4 seconds, with division occurring about 1 mm from the LIMA wall to minimise damage to the LIMA conduits. In both harvesting techniques, the LIMA was dissected proximally above the first intercostal branch and distally till bifurcation into the musculophrenic and superior epigastric arteries while avoiding coagulation of adjacent veins.

After 2–3 minutes of systematic heparinisation, achieving an ACT of >350 seconds, the distal end of the LIMA was divided at or just before the level of the bifurcation. Once the conduit was divided, LIMA flow quality was assessed visually by allowing the graft to bleed for a few seconds. The distal end was then gently clamped with an atraumatic bulldog clamp or clipped to prevent unnecessary bleeding while we prepared for distal anastomosis. We applied intermittent topical warm papaverine spray throughout the LIMA harvest, maintaining systolic blood pressure around 100 mmHg. After harvesting, the LIMA was wrapped with warm papaverine-soaked gauze, avoiding direct papaverine injection into the lumen to prevent damage, until it's needed to improve the spasm. Care was taken to prevent phrenic nerve injury during proximal dissection. Further, direct grasping of the LIMA with forceps was avoided by grasping nearby adventitial tissue. In both cases, we opened the left pleura to enhance LIMA visualisation and placed a drain tube as a precaution against cardiac tamponade – a standard practice in our procedure; though postoperative chest reopening due to bleeding was rare, occurring in less than 0.5% of our cases. Both the skeletonised and semi-skeletonised LIMA harvesting techniques were performed by two equally qualified surgeons, those who had completed postgraduate surgical training in cardiac surgery under a single-surgeon practice and performed both harvesting techniques stochastically. We used the same standard

LIMA harvesting techniques for both elective and emergency CABG surgeries, as well as for redo CABG procedures when the LIMA is available for harvesting. Furthermore, we employed reusable Ligaclips applicators to apply the Ligaclips during these harvesting procedures.

## Statistical analysis

We utilised version 28.0 SPSS (Statistical Package for the Social Sciences) software for statistical analysis. Initially, a univariate analysis was conducted to ascertain the difference patterns in the BANGABANDHU dataset. Study variables with a p value of ≤0.05 between LIMA harvesting techniques were included in a multivariate logistic regression model to identify the independent variables associated with the benefits of semi-skeletonised LIMA. Additionally, the goodness-of-fit of the prediction model was confirmed by the area under the receiver operating characteristic (AUROC) curve. A p-value of <0.05 was considered statistically significant.

## Results

The BANGABANDHU study evaluated 2209 age- and sex-matched adult Bangladeshi isolated CABG patients, which included 1050 patients with a skeletonised LIMA and 1159 patients with a semi-skeletonised LIMA. Preoperative risk profiles and comorbidities (EuroSCORE II, hypertension, diabetes, renal impairment, COPD, left main disease and multivessel CAD) were identical between study groups (p>0.05) (Table 1). However, LIMA length (15.7±0.3 vs 15.5±0.5; p=0.05), harvest time in minutes (35.9±5.5 vs 16.6±3.9; p<0.001) and number of used Ligaclip (23.6±4.8 vs 11.7±3.6; p<0.001) were significantly higher in skeletonised compared to the semi-skeletonised LIMA sample. Additionally, postoperative pain and superficial or deep sternal wound infections were similar between study groups.

Table 1. Baseline characteristics of study population (n=2209).

| Variables | Skeletonised LIMA (n=1050) | Semi-skeletonised LIMA (n=1159) | P value |
|---|---|---|---|
| Age (mean ±SD) | 57.3±6.9 | 57.9±7.1 | 0.09 |
| Male | 899 (85.7%) | 999 (86.2%) | 0.81 |
| Preop EuroSCORE II | 3.8±0.8 | 3.9±0.9 | 0.07 |
| Hypertension | 921 (87.7%) | 1041 (89.8%) | 0.21 |
| Diabetes mellitus | 780 (74.3%) | 887 (76.9%) | 0.27 |
| Reanl impairment | 23 (2.2%) | 12 (1.0%) | 0.07 |
| COPD | 113 (10.8%) | 117 (10.1%) | 0.67 |
| LM disease | 43 (4.1%) | 42 (3.6%) | 0.64 |
| Multivessel CAD | 950 (90.5%) | 1037 (89.5%) | 0.53 |
| LVEF <30% | 72 (6.9%) | 94 (8.1%) | 0.41 |
| Emergency CABG | 18 (1.7%) | 29 (2.5%) | 0.34 |
| **Operative variables** | | | |
| LIMA length (cm) (mean ±SD) | 15.7±0.3 | 15.5±0.5 | 0.05 |
| LIMA harvest time (minute) | 35.9±5.5 | 16.6±3.9 | <0.001 |
| Number of LigaClip (mean ±SD) | 23.6±4.8 | 11.7±3.6 | <0.001 |
| LIMA spasm | 16 (1.5%) | 8 (0.7%) | 0.11 |
| LIMA haematoma | 12 (1.1%) | 0 (0.0%) | <0.001 |
| Extra-pleural LIMA | 723 (68.9%) | 746 (64.4%) | 0.08 |
| Re-exploration | 16 (1.5%) | 20 (1.7%) | 0.76 |

Here, COPD -chronic obstructive pulmonary disease; CAD -coronary artery disease; LVEF -left ventricular ejection fraction; and LIMA -left internal mammary artery. The P-value reached from the chi-square test and independent t-test, as appropriate; P<0.05 confirms statistical significance.

An age and gender adjusted multivariate logistic regression model found LIMA harvest time (odds ratio [OR] 0.067, 95% CI 0.01–0.39; p = 0.003) and number of Ligaclips used (OR 0.561, 95% CI 0.41–0.76; p < 0.001) were significantly and positively associated with the semi-skeletonised LIMA techniques (Table 2). The OR demonstrated that the harvest time and the number of used Ligaclips were 93.3% and 43.9% lower, respectively, in the semi-skeletonised LIMA compared to the skeletonised LIMA. The area under the receiver operating characteristic (AUROC) curve was 0.99 for the LIMA harvest time and 0.97 for the number of Ligaclips used, representing the goodness-of-fit of the model (Fig 2). Furthermore, the LIMA harvest time showed a sensitivity of 98.5% and specificity of 88.5%, while the number of Ligaclips used had a sensitivity of 92.5% and specificity of 82.0% among semi-skeletonised LIMA technique samples. Furthermore, overall LR model quality was excellent 99.9% for LIMA harvest time and 97.5% for the number of Ligaclip used variables in predicting advantage of semi-skeletonised LIMA techniques (S1 Fig).

## Discussion

We found that the LIMA harvesting time and number of Ligaclips used have significantly lower odds, approximately 93.3% and 43.9%, respectively, among the semi-skeletonised LIMA group compared to the skeletonised LIMA group. Further, identical sociodemographic variables and complications associated with LIMA harvesting techniques validate our findings on the advantage of semi-skeletonised over skeletonised LIMA techniques.

**Table 2. Age and gender adjusted multivariate logistic regression model predicting advantages of semi-skeletonised LIMA techniques.**

|  | Odds Ratio | 95% CI | | P value |
|---|---|---|---|---|
|  |  | Lower | Upper |  |
| LIMA harvest time (min) | 0.067 | 0.01 | 0.39 | 0.003 |
| Ligaclip used (number) | 0.561 | 0.41 | 0.76 | <0.001 |
| LIMA length (cm) | 0.191 | 0.01 | 11.10 | 0.42 |
| Age | 0.949 | 0.84 | 1.07 | 0.40 |
| Gender | 0.067 | 0.01 | 12.57 | 0.31 |

Dependent variable: Semi-skeletonised LIMA techniques.

Variable(s): LIMA length in cm, LIMA harvest time, Ligaclip used, Age, Gender. CI -Confidence Interval.

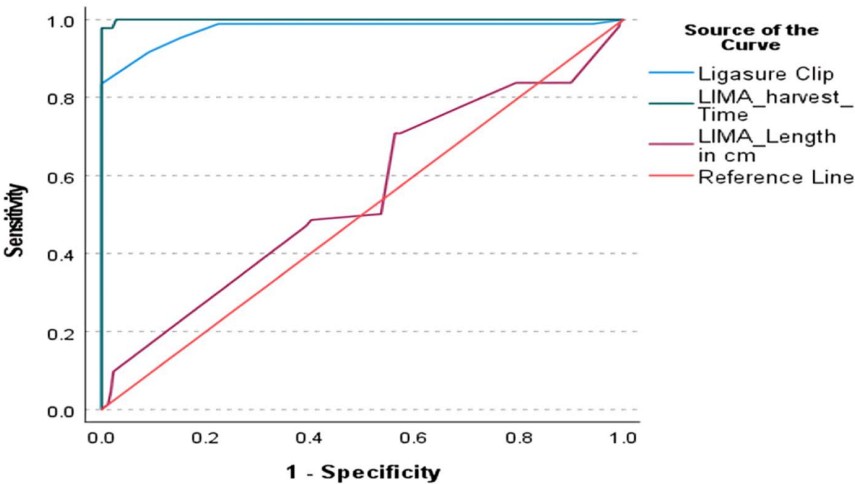

**Fig 2. The receiver operating characteristic (ROC) curve illustrates the area under the ROC curve with the sensitivity and specificity of the risk prediction model.**

Skeletonised LIMA harvesting typically takes longer than pedicle techniques due to the meticulous dissection of the surrounding tissues to avoid trauma to the LIMA, which aims to provide better conduit quality [3–8,18]. Despite similar surgical skills, we found that semi-skeletonised LIMA harvesting is faster due to simpler dissection, which preserves some surrounding tissues, requiring less meticulous dissection and fewer clips of small LIMA branches, resulting in quicker conduit preparation compared to skeletonised LIMA techniques. Additionally, the higher skeletonised LIMA harvesting time results in an extended operative time, which is associated with increased resource utilisation, such as operating room time, anaesthesia services, and staffing, leading to increased operative costs [19,20]. Depending on the surgeon's experience, the average skeletonised LIMA harvest time ranges from 25 to 45 minutes, similar to our findings [3–8,10,13]. Further, existing literature found that skeletonised LIMA was associated with lower wound infections, postoperative pain, short hospital stays, and better preservation of lung function but required longer operative time [4,6]. Contrarily, studies also observed that skeletonised LIMA has no early or mid-term outcomes benefits, [7] and does not appear to lower the occurrence of postoperative chest pain despite causing significantly less inner chest wall trauma [8]. Furthermore, higher operative time substantially impacts the total costs of CABG procedures, increasing hospital expenses and the risk of postoperative complications, [20] which supports less time-consuming but safe semi-skeletonised LIMA harvest techniques.

Skeletonised LIMA harvesting required more extensive use of Ligaclips to secure the numerous small branches of the artery than the pedicled method [21]. In another study, Wendler et al. [22] indicated that the skeletonised technique, by exposing more arterial branches, indeed necessitated a greater number of Ligaclips to achieve haemostasis. Additionally, several published papers also stated that with experienced surgical techniques, the increased use of Ligaclips can be minimised even with skeletonised LIMA harvesting, which implies that the number of Ligaclips used may be more related to the surgeon's proficiency and technique rather than the harvesting method alone [23–25]. Furthermore, a recent Randomised Control Trial found that a semi-skeletonised LIMA graft to the left anterior descending artery resulted in good distal run-off with less operative time, supporting our study findings [26]. Despite similar LIMA spasm and re-exploration events, we found the LIMA haematoma rate was significantly lower in the semi-skeletonised LIMA group, which also supports the safety profile of the semi-skeletonised technique, consistent with existing study findings [11,13,27]. This is the first South Asian study to observe that semi-skeletonised LIMA was not inferior to skeletonised LIMA in terms of number of used Ligaclips, harvest time and early postoperative bleeding during off-pump CABG surgery.

## Strength and limitation

Despite the robustness of the study results, a few limitations must be acknowledged, especially the observational nature of the study. The study population is primarily of Bangladeshi ancestry, so the LIMA length might not be generalised to other Asian countries or across the globe. Despite ongoing debate regarding the optimal length of the LIMA in various harvesting techniques, our protocol specified dissection of the LIMA from proximal to the first intercostal branch, extending to the distal bifurcation into the musculophrenic and superior epigastric arteries. This approach was designed to minimize outcome bias related to conduits length and harvesting time. The harvesting time and number of Ligaclip used might vary between surgical trainees and surgeons; however, utilising a database from all equally qualified surgeons who completed postgraduate surgical training in cardiac surgery in a single surgeon practice minimises the risk of outcome bias. Nevertheless, further studies evaluating the cost-effectiveness of semi-skeletonised versus skeletonised LIMA, considering factors such as total procedural costs, length of hospital stay, and follow-up expenses, may shed more light on the current study findings. Additionally, we acknowledge the potential for recruitment bias due to the ambispective nature of the study; however, using an age- and sex-matched study sample with similar comorbidities and standardised data collection tools minimises this risk of bias. While the mean difference in LIMA length between study groups is very minimal, a p-value of 0.05 raises concerns about a type I error. However, a large sample size with low variability in LIMA length can make even minor differences more detectable, leading to potential misinterpretation [28]. The inclusion of all primary, isolated elective, and emergency CABG, as well as redo CABG cases with available LIMA, may raise concerns about bias in harvesting

techniques and the number of Ligaclip uses. However, the harvesting techniques for the LIMA remained consistent, which helps mitigate the risk of bias. Finally, utilising similar clinical profiles of study participants cross-validates our short harvesting time, representing lower total operative time, and the low number of used Ligaclips, which demonstrates the resource utilisation efficiency of the semi-skeletonised LIMA harvest technique.

## Conclusion

Semi-skeletonised techniques are safe and feasible, with resource utilisation efficiency on LIMA harvesting, especially significantly shorter harvesting time and fewer Ligaclips than skeletonised techniques. We recommend conducting further studies to assess the long-term graft patency and survival benefits of semi-skeletonised compared to skeletonised LIMA in CABG surgery to validate the robustness of the current study's findings.

## Suppvorting information

**S1 Fig. The goodness-of-fit of the logistic regression model.**
(DOCX)

## Author contributions

**Conceptualization:** Redoy Ranjan, Aziz Momin, Riyaz A. Kaba, Gie Ken-Dror, Sanjay Kumar Raha, Md Kamrul Hasan, Venkatachalam Chandrasekaran, Asit Baran Adhikary.

**Data curation:** Redoy Ranjan, Asit Baran Adhikary.

**Formal analysis:** Redoy Ranjan, Aziz Momin, Riyaz A. Kaba, Gie Ken-Dror, Sanjay Kumar Raha, Md Kamrul Hasan, Venkatachalam Chandrasekaran, Asit Baran Adhikary.

**Investigation:** Redoy Ranjan, Asit Baran Adhikary.

**Methodology:** Redoy Ranjan, Aziz Momin, Riyaz A. Kaba, Gie Ken-Dror, Sanjay Kumar Raha, Md Kamrul Hasan, Venkatachalam Chandrasekaran, Asit Baran Adhikary.

**Resources:** Redoy Ranjan, Aziz Momin, Riyaz A. Kaba, Gie Ken-Dror, Sanjay Kumar Raha, Md Kamrul Hasan, Venkatachalam Chandrasekaran, Asit Baran Adhikary.

**Supervision:** Aziz Momin, Riyaz A. Kaba, Gie Ken-Dror, Sanjay Kumar Raha, Md Kamrul Hasan, Venkatachalam Chandrasekaran, Asit Baran Adhikary.

**Validation:** Aziz Momin, Riyaz A. Kaba, Gie Ken-Dror, Sanjay Kumar Raha, Md Kamrul Hasan, Venkatachalam Chandrasekaran, Asit Baran Adhikary.

**Visualization:** Redoy Ranjan, Riyaz A. Kaba, Gie Ken-Dror, Asit Baran Adhikary.

**Writing – original draft:** Redoy Ranjan.

**Writing – review & editing:** Redoy Ranjan, Aziz Momin, Riyaz A. Kaba, Gie Ken-Dror, Sanjay Kumar Raha, Md Kamrul Hasan, Venkatachalam Chandrasekaran, Asit Baran Adhikary.

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
