## [Decision Letter · Decision Letter 0]

12 Mar 2025

Dear Dr. Ranjan,

Thank you for submitting your manuscript to PLOS ONE. After careful consideration, we feel that it has merit but does not fully meet PLOS ONE’s publication criteria as it currently stands. Therefore, we invite you to submit a revised version of the manuscript that addresses the points raised during the review process.

We look forward to receiving your revised manuscript.

Kind regards,

Usama Waqar, M.B.B.S

Academic Editor

PLOS ONE

Reviewers' comments:

Reviewer's Responses to Questions

**Comments to the Author**

1. Is the manuscript technically sound, and do the data support the conclusions?

Reviewer #1: Yes

Reviewer #2: Partly

Reviewer #3: Partly

2. Has the statistical analysis been performed appropriately and rigorously?

Reviewer #1: I Don't Know

Reviewer #2: Yes

Reviewer #3: Yes

3. Have the authors made all data underlying the findings in their manuscript fully available?

Reviewer #1: No

Reviewer #2: Yes

Reviewer #3: Yes

4. Is the manuscript presented in an intelligible fashion and written in standard English?

Reviewer #1: Yes

Reviewer #2: Yes

Reviewer #3: Yes

Reviewer #1: PLOS ONE

Safety and cost-effectiveness of semi-skeletonised versus skeletonised left internal mammary artery harvesting techniques: the BANGABANDHU study

PONE-D-25-07678

March 7, 2025

I want to thank the authors for their research efforts and for the opportunity to review their manuscript.

I have included some comments below followed by some line items.

First, the three techniques would be skeletonised, semi-skeletonised and pedicled. Based on your description of semi-skeletonised, you need to explain how that is different than a pedicled LIMA (a pedicle also includes the veins, adipose, endothoracic fascia and muscle). I cannot tell the difference between semi-skeletonised and pedicled other than some arbitrary distance from the LIMA that is included in the dissection.

Second, you mention the length of the LIMA as a difference but you say that the protocol requires the same distance from proximal to the first intercostal branch to the bifurcation, so how is there a difference in length? In the results, the lengths need to be included. The difference was found to be significant at p=0.05 with the difference being 0.2 (which is inside the standard deviation so I am not sure how that is statistically significant). This seems to not be a important difference.

Third, there is more than a doubling of time between skeletonised and semi-skeletonised. That seems to be excessive. This should be explored or explained a little more.

Fourth, you should make clear how many surgeons were included in this study. I believe you are saying that they all trained under the same surgeon. Is the current population from a single surgeon practice? Is it the CABG surgeon taking down the IMA or is it an assistant surgeon? This area of how many surgeons are participating is important to document.

Line 36: “that evaluated 2209 age- and sex-matched adult Bangladeshi isolated CABG patients”

Line 38: I am not sure “curtailed” is the correct word here

Line 39: should be consistent and use “-“ in semi-skeletonised or not (this is the first time it was not used)

Line 42: “we evaluated 2209 age- and sex-matched adult isolated CABG patients”

Line 44: should not capitalize anything other than EuroSCORE II and COPD

Line 45: usually, “p” is not capitalized (p> 0.05)

Line 48: Further more, AN age- and sex-adjusted multivariate….

Line 52: you mention cost-effectiveness here without any discussion of it in the Results (you should not include a result in the Conclusion that was not presented in the Results)

Line 86: “its blood supply” – do you mean the chest wall’s blood supply? It is not clear in the way this sentence is written (the blood supply presumably refers to the chest wall and minimal surrounding tissue presumably refers to the LIMA but these two items are separated by an “and” which presumes both items are related to the same thing).

Line 95: “which is expensive” – I would say costs more money, but “expensive” is relative and in comparison with the cost of the entire operation, the Ligaclip is not a major driver of cost.

Line 51: “We recruited 2209 age- and sex-matched adult CABG patients”

Line 53: “We divided the study population into 2 groups based on….”

Line 65: no need for the word “specifically”

Line 79: when skeltonising, were Ligasure clips only used on the LIMA side just like in the semi-skeletonised scenario? This would be a difference in treatment if they were not the same.

Line 86: “conduit is” – the paper has been written in the past tense, you switch to present here and then continue in the present tense in the next sentence.

Line 93: “direct grasping of the LIMA with forceps was avoided by grasping nearby adventitial tissue.”

Line 208: you switch from preterite to present here as well

Line 223: “evaluated 2209 age- and sex-matched adult Bangladeshi isolated CABG patients which included 1050 with a skeletonized LIMA and 1159 with a semi-skeletonised LIMA.”

Line 225: no words should be capitalized other than EuroSCORE II, COPD and CAD)

Line 227: you should include the lengths, times and number of clips here

Line 228: “number of Ligaclips used”

Line 228: “compared to THE semi-skeletonised”

Line 231: AN age- and gender-adjusted

Line 232: “number of Ligaclips used”

Line 233: were significantly and positively associated with the semi-skeletonised LIMA technique

Line 234-236: I do not understand this sentence

Line 236: this sentence needs to be re-worked

Line 284: this sentence needs to be re-worked

Line 288: “is not”

Line 297: if the protocol was from proximal to the first intercostal branch to the bifurcation, then why was the skeletonised longer if both were prepared the same way in terms of length?

Line 302: I am not sure that this “mitigates the risk of outcome bias”

Reviewer #2: This manuscript addresses a clinically relevant topic comparing semi-skeletonised and skeletonised LIMA harvesting techniques. However, several key issues limit its current suitability for publication.

First, the authors emphasize cost-effectiveness but provide inadequate economic analysis. Simply counting Ligaclips is insufficient; a detailed cost breakdown including operative time, resource utilization, and hospital stay should be presented.

Second, the multivariate logistic regression model raises concerns. The odds ratios reported (e.g., OR 10.8 for LIMA harvest time) appear unusually large, suggesting potential errors in statistical modeling or interpretation. Clarification and re-analysis are needed.

Third, the authors mention early complications but do not adequately define or systematically evaluate them. Clear definitions, standardized follow-up periods, and explicit complication rates should be provided.

Lastly, the manuscript lacks clarity regarding surgeon experience standardization. Although a single surgeon's practice is mentioned, variability among trainees and junior surgeons may significantly confound results.

Reviewer #3: Dear Authors,

Thank you for submitting your manuscript for review. I was pleased to receive it.

Your study addresses a clinically relevant question and provides useful insights into the practical and cost-effective aspects of IMA harvesting techniques. However, I have some comments for your consideration:

1. In the abstract conclusion, you could briefly include clinical relevance or outcomes (e.g., complication rates, recovery, postoperative morbidity) alongside procedural efficiency.

2. You mention the “ambispective” nature of the study. Could you clarify how the prospective and retrospective data were integrated and managed to avoid biases regarding quality and completeness across the two collection approaches?

3. Why were certain covariates (e.g., hypertension, diabetes, COPD) matched but not directly included in the multivariate regression model? Could you explain this choice in the Methods?

4. Regarding the harvesting techniques, descriptions are clear. However, could you include additional intraoperative images (if available) to show differences between the techniques?

5. The use of Ligaclips as a cost indicator is well-justified. Still, could you provide an economic analysis or at least a brief comparative analysis of the overall economic impact (e.g., average procedural cost differences or estimated financial savings per case)?

6. Your confidence intervals in the logistic regression (e.g., OR 10.8 with CI 0.01-0.39) seem incorrect in terms of directionality. Typically, OR >1 indicates increased odds. Please clarify.

7. Can you compare semi-skeletonised and skeletonised techniques regarding long-term patency or graft survival, if this data is available?

8. Given that all surgeries were performed by qualified surgeons within a single surgeon's practice, how generalizable do you believe these results are? Could you briefly discuss how the familiarity of surgeons with each technique may influence results?

9. You have mentioned the identical comorbidities between study groups, demonstrating good baseline comparability. However, given the single-center, single-surgeon practice setting, could there be any institutional protocols or perioperative management peculiarities that may have influenced outcomes differently from broader multi-institutional contexts?

Thank you again for the opportunity to review your work.

**Do you want your identity to be public for this peer review?** For information about this choice, including consent withdrawal, please see our Privacy Policy

Reviewer #1: No

Reviewer #2: **Yes: ** Robert J. Chen, MD, MPH

Reviewer #3: **Yes: ** Savvas Lampridis

---

## [Author Response · Author response to Decision Letter 1]

28 Mar 2025

Reviewer #1: I want to thank the authors for their research efforts and for the opportunity to review their manuscript. I have included some comments below followed by some line items.

Q: First, the three techniques would be skeletonised, semi-skeletonised and pedicled. Based on your description of semi-skeletonised, you need to explain how that is different than a pedicled LIMA (a pedicle also includes the veins, adipose, endothoracic fascia and muscle). I cannot tell the difference between semi-skeletonised and pedicled other than some arbitrary distance from the LIMA that is included in the dissection.

Response: I appreciate your concerns, which are not untrue. However, as a cardiac surgeon, you must recognise that minimising tissue damage during LIMA harvesting is crucial, especially since pedicled LIMA grafts are prone to sternal wound infections, particularly in patients with comorbidities. Therefore, we hypothesised that harvesting in a semi-skeletonised fashion, as you said, "some arbitrary distance from the LIMA that is included in the dissection", could be safe. Despite the lack of a standard definition, we defined the semiskelitonised technique in our study on page 7, lines 181-183 (yellow highlighted), which might be helpful in future studies to define semi-skeletonised LIMA.

Q: Second, you mention the length of the LIMA as a difference but you say that the protocol requires the same distance from proximal to the first intercostal branch to the bifurcation, so how is there a difference in length? In the results, the lengths need to be included. The difference was found to be significant at p=0.05 with the difference being 0.2 (which is inside the standard deviation so I am not sure how that is statistically significant). This seems to not be a important difference.

Response: Yes, we agreed, and in this study, we set a p-value of <0.05 as statistically significant, which we mentioned on page 8, lines 216,217 (yellow highlighted).

Q: Third, there is more than a doubling of time between skeletonised and semi-skeletonised. That seems to be excessive. This should be explored or explained a little more.

Response: While we mentioned it on page 10, lines 257-259 (highlighted in yellow), we have also added a brief explanation in lines 260-262 (highlighted in yellow).

Q: Fourth, you should make clear how many surgeons were included in this study. I believe you are saying that they all trained under the same surgeon. Is the current population from a single surgeon practice? Is it the CABG surgeon taking down the IMA or is it an assistant surgeon? This area of how many surgeons are participating is important to document.

Response: All study patients are from a single surgeon's practice, which we mentioned on page 6, lines 152,153. Further, the LIMA was harvested by two equally trained cardiac surgeons, which we mentioned in methodology sections lines 204-207.

Q: Line 36: "that evaluated 2209 age- and sex-matched adult Bangladeshi isolated CABG patients"

Response: Modifications are done as you suggested

Q: Line 38: I am not sure "curtailed" is the correct word here

Response: Thanks for your suggestions, "curtailed" replaced with "observed".

Q: Line 39: should be consistent and use "-"in semi-skeletonised or not (this is the first time it was not used)

Response: Modifications are done as you suggested

Q: Line 42: "we evaluated 2209 age- and sex-matched adult isolated CABG patients"

Response: Modifications are done as you suggested

Q: Line 44: should not capitalize anything other than EuroSCORE II and COPD

Response: Modifications are done as you suggested

Q: Line 45: usually, "p" is not capitalized (p> 0.05)

Response: Modifications are done as you suggested throughout the manuscript.

Q: Line 48: Furthermore, AN age- and sex-adjusted multivariate….

Response: Modifications are done as you suggested

Q: Line 52: you mention cost-effectiveness here without any discussion of it in the Results (you should not include a result in the Conclusion that was not presented in the Results)

Response: Conclusion modified as you suggested

Q: Line 86: "its blood supply" – do you mean the chest wall's blood supply? It is not clear in the way this sentence is written (the blood supply presumably refers to the chest wall and minimal surrounding tissue presumably refers to the LIMA but these two items are separated by an "and" which presumes both items are related to the same thing).

Response: The statement was modified as you suggested, with lines 84-86 (yellow highlighted).

Q: Line 95: "which is expensive" – I would say costs more money, but "expensive" is relative and in comparison with the cost of the entire operation, the Ligaclip is not a major driver of cost.

Response: Statement modified as you suggested, line 95 (yellow highlighted).

Q: Line 151: "We recruited 2209 age- and sex-matched adult CABG patients"

Response: Statement modified as you suggested, yellow highlighted

Q: Line 153: "We divided the study population into 2 groups based on…."

Response: Statement modified as you suggested, yellow highlighted

Q: Line 165: no need for the word "specifically"

Response: Statement modified as you suggested, yellow highlighted

Q: Line 179: when skeltonising, were Ligasure clips only used on the LIMA side just like in the semi-skeletonised scenario? This would be a difference in treatment if they were not the same.

Response: Yes. Ligaclips were usually used on the LIMA side in both the semi-skeletonised & skeletonised techniques which we mentioned in lines 179-181 (yellow highlighted).

Q: Line 186: "conduit is" – the paper has been written in the past tense, you switch to present here and then continue in the present tense in the next sentence.

Response: I appreciate your concern; line 193 (yellow highlighted) has been addressed and corrected.

Q: Line 193: "direct grasping of the LIMA with forceps was avoided by grasping nearby adventitial tissue."

Response: Modifications are done as you suggested, line 200 (yellow highlighted).

Q: Line 208: you switch from preterite to present here as well

Response: Modifications are done as you suggested, line 217, (yellow highlighted).

Q: Line 223: "evaluated 2209 age- and sex-matched adult Bangladeshi isolated CABG patients which included 1050 with a skeletonized LIMA and 1159 with a semi-skeletonised LIMA."

Response: Modifications are done as you suggested, lines 224-226 (yellow highlighted).

Q: Line 225: no words should be capitalized other than EuroSCORE II, COPD and CAD)

Response: Modifications are done as you suggested

Q: Line 227: you should include the lengths, times and number of clips here

Response: Modifications are done as you suggested, lines 228-230 (yellow highlighted).

Q: Line 228: "number of Ligaclips used"

Response: Modifications are done as you suggested

Q: Line 228: "compared to THE semi-skeletonised"

Response: Modifications are done as you suggested, line 231 (yellow highlighted).

Q: Line 231: AN age- and gender-adjusted

Response: Modifications are done as you suggested, line 233 (yellow highlighted).

Q: Line 232: "number of Ligaclips used"

Response: Modifications are done as you suggested

Q: Line 233: were significantly and positively associated with the semi-skeletonised LIMA technique

Response: Modifications are done as you suggested

Q: Line 234-236: I do not understand this sentence

Response: Sorry for the inconvenience. We have updated the statement as you suggested, and the changes are highlighted in yellow, lines 238-243.

Q: Line 236: this sentence needs to be re-worked'

Response: Statement modified as you suggested, yellow highlighted

Q: Line 284: this sentence needs to be re-worked

Response: Statement modified, lines 283-285 (yellow highlighted).

Q: Line 288: "is not"

Response: Modifications are done as you suggested, line 286 (yellow highlighted).

Q: Line 297: if the protocol was from proximal to the first intercostal branch to the bifurcation, then why was the skeletonised longer if both were prepared the same way in terms of length?

Response: We understand your concerns; however, the LIMA length was not significantly different between the study groups. This study was conducted because our senior consultants believe that a skeletonised LIMA provides more length, which was not the scenario in our study. However, the minimum length difference may be due to the adjacent fascia and veins associated with a semi-skeletonised LIMA rather than a skeletonised one.

Q: Line 302: I am not sure that this "mitigates the risk of outcome bias"

Response: We understand your concerns; however, every study has the potential for risk of bias, and we did our best to minimise the risk of bias by utilising two equally qualified surgeons to harvest the LIMA. Additionally, we changed "mitigates" to "minimises" in line 301 (highlighted in yellow).

Reviewer #2: This manuscript addresses a clinically relevant topic comparing semi-skeletonised and skeletonised LIMA harvesting techniques. However, several key issues limit its current suitability for publication.

Q: First, the authors emphasize cost-effectiveness but provide inadequate economic analysis. Simply counting Ligaclips is insufficient; a detailed cost breakdown including operative time, resource utilization, and hospital stay should be presented.

Response: We appreciate your concerns. The current study aimed to evaluate specific research questions regarding the safety and effectiveness of the LIMA harvesting technique rather than the whole CABG procedure, including hospital stay. Nevertheless, the overall operation cost was similar as all cases were performed by the same surgeon using off-pump techniques. However, we have revised our study objective statement for better clarity for our readers on page 2, and lines 31-33 (yellow highlighted).

Q: Second, the multivariate logistic regression model raises concerns. The odds ratios reported (e.g., OR 10.8 for LIMA harvest time) appear unusually large, suggesting potential errors in statistical modeling or interpretation. Clarification and re-analysis are needed.

Response: We agreed, and thank you very much for your suggestions. We revised the LR model by considering semi-skeletonised LIMA as a dependent variable. The modified OR values replaced the previous one on page 9, including an interpretation of our findings, lines 234-238 (yellow highlighted).

Q: Third, the authors mention early complications but do not adequately define or systematically evaluate them. Clear definitions, standardized follow-up periods, and explicit complication rates should be provided.

Response: This study evaluated specific research questions related to the intra-operative LIMA harvesting technique rather than the whole CABG procedure and follow-up outcomes. However, we modified the definition of study variables, including complications on page 6 and lines 168-172 (yellow highlighted).

Q: Lastly, the manuscript lacks clarity regarding surgeon experience standardization. Although a single surgeon's practice is mentioned, variability among trainees and junior surgeons may significantly confound results.

Response: Two equally qualified surgeons harvested LIMA who had completed postgraduate surgical training in cardiac surgery under a single-surgeon practice, which is mentioned on page 8 and lines 204-207 (yellow highlighted). However, we agreed that there might be a bias between surgeons, which we acknowledged as study limitations on page 11/12, lines 299-302 (yellow highlighted).

Reviewer #3: Dear Authors, Thank you for submitting your manuscript for review. I was pleased to receive it. Your study addresses a clinically relevant question and provides useful insights into the practical and cost-effective aspects of IMA harvesting techniques. However, I have some comments for your consideration:

1. In the abstract conclusion, you could briefly include clinical relevance or outcomes (e.g., complication rates, recovery, postoperative morbidity) alongside procedural efficiency.

Response: Thanks for your suggestion; however, an abstract conclusion should ideally be based on the most significant findings, which are short and precise. As you suggested, we modified our conclusive statement on page 2 (yellow highlighted).

2. You mention the "ambispective" nature of the study. Could you clarify how the prospective and retrospective data were integrated and managed to avoid biases regarding quality and completeness across the two collection approaches?

Response: We detailed the "ambispective" nature of the study on page 6, lines 158-167 (highlighted in yellow). Further, we have added a statement on how we integrated and managed quality and completeness across the two data collection approaches on page 6 and acknowledged a limitations statement on the risk of potential bias on page 12, lines 302-304 (yellow highlighted).

3. Why were certain covariates (e.g., hypertension, diabetes, COPD) matched but not directly included in the multivariate regression model? Could you explain this choice in the Methods?

Response: We detailed the selection of study variables for the regression model on page 8, lines 212-215 (highlighted in yellow).

4. Regarding the harvesting techniques, descriptions are clear. However, could you include additional intraoperative images (if available) to show differences between the techniques?

Response: Thanks, I appreciate your comments. We have added more pictures of skeletonised & semi-skeletonised LIMA, and Figure 1 has been modified as you suggested.

5. The use of Ligaclips as a cost indicator is well-justified. Still, could you provide an economic analysis or at least a brief comparative analysis of the overall economic impact (e.g., average procedural cost differences or estimated financial savings per case)?

Response: We appreciate your concerns. The current study aimed to evaluate specific research questions regarding the safety and effectiveness of the LIMA harvesting technique rather than the whole CABG procedure, including hospital stay. Nevertheless, the overall operation cost was similar as all cases were performed by the same surgeon using off-pump techniques. However, we have revised our study objective statement for better clarity for our readers on page 2, and lines 31-33 (yellow highlighted).

6. Your confidence intervals in the logistic regression (e.g., OR 10.8 with CI 0.01-0.39) seem incorrect in terms of directionality. Typically, OR >1 indicates increased odds. Please clarify.

Response: We agreed, and thank you very much for your suggestions. We revised the LR model by considering semi-skeletonised LIMA as a dependent variable. The modified OR values replaced the previous one on page 9, including an interpretation of our findings, lines 234-238 (yellow highlighted).

7. Can you compare semi-skeletonised and skeletonised techniques regarding long-term patency or graft survival, if this data is available?

Response: We apologise, but this study focuses specifically on evaluating the LIMA harvesting technique; unfortunately, we do not have long-term follow-up data on patency or graft survival. However, in response to your suggestions, we have included a recommendation for future studies to validate the robustness of our findings on page 12, lines 314-316 (yellow highlighted).

8. Given that all surgeries were performed by qualified surgeons within a single surgeon's practice, how generalizable do you believe these results are? Could you briefly discuss how the familiarity of surgeons with each technique may influence results?

Response: We understand your concerns; however, Professor Adhikary is one of the most senior CTh surgeons, with over 30 years of experience in CABG surgery. He has trained surgeons at the only medical university hospital in Bangladesh, which is why we believe the outcome results are generalisable despite the inherent risk of bias that is standard in any observational study. Additionally, in this study, the LIMA was harvested by two equally trained cardiac surgeons, as noted on page 8, lines 204-207 (highlighted in yellow). However, we agreed that there might be a bias between surgeons, which we acknowledged as study limitations on page 11/12, lines 299-3

---

## [Decision Letter · Decision Letter 1]

7 May 2025

Dear Dr. Ranjan,

We look forward to receiving your revised manuscript.

Kind regards,

Usama Waqar, M.B.B.S

Academic Editor

PLOS ONE

Reviewers' comments:

Reviewer's Responses to Questions

**Comments to the Author**

Reviewer #1: All comments have been addressed

Reviewer #2: (No Response)

Reviewer #3: (No Response)

2. Is the manuscript technically sound, and do the data support the conclusions?

Reviewer #1: Yes

Reviewer #2: Partly

Reviewer #3: Partly

3. Has the statistical analysis been performed appropriately and rigorously?

Reviewer #1: I Don't Know

Reviewer #2: Yes

Reviewer #3: Yes

4. Have the authors made all data underlying the findings in their manuscript fully available?

Reviewer #1: Yes

Reviewer #2: Yes

Reviewer #3: Yes

5. Is the manuscript presented in an intelligible fashion and written in standard English?

Reviewer #1: Yes

Reviewer #2: Yes

Reviewer #3: Yes

Reviewer #1: PLOS ONE

Safety and cost-effectiveness of semi-skeletonised versus skeletonised left internal mammary artery harvesting techniques: the BANGABANDHU study

PONE-D-25-07678R1

April 17, 2025

I want to thank the authors for their thorough responses to prior comments.

I have one comment below followed by some line items.

1. I am still concerned that the central item around which the entire paper revolves is the “semi-skeletonized” LIMA and you are unable to provide a definition that distinguishes it from the only other most commonly used harvest technique which is the pedicled technique. The two most common techniques are skeletonized and pedicled and so it is glaring that this paper does not mention the pedicled technique in any substantial or comparative way. In order to do that, you would have to distinguish the semi-skeletonized and the pedicled and I do not know how you would distinguish those. As you mention, as a cardiac surgeon, they all recognize that minimizing tissue damage during LIMA harvesting is crucial so pedicled LIMAs are usually small (surgeons shy away from taking a large swath of chest wall when taking the pedicle) so how small is small to qualify as semi-skeletonized as opposed to a small pedicle? I think you need to try an attempt at distinguishing the semi-skeletonized vs a pedicled LIMA since even though you do not bring up pedicled LIMAs (other than a few sentences in the Discussion), every surgeon reading this will be wondering what makes your technique different than theirs (if they take small pedicles).

Line 57-60: I am not sure what section this belongs to, but there needs to be a reference noted for all these claims

Line 102: to what drainage are you referring?

Line 154-155: this should be mentioned in limitations that you divided (did not randomize) the group into how the LIMA was harvested. It is possible that a significant sorting mechanism was already operating when the patient was chosen for skeletonized vs semi-skeletonized.

Line 180: the word “should” should not be here – you are describing what happened, you are not explaining how to do it

Line 280: you are now comparing semi-skeletonized to pedicled – I am not sure of the relevance to this paper as you have not discussed pedicled LIMAs. What is the drainage compared to the skeletonized (the comparison in this paper)?

Reviewer #2: The authors' revisions significantly improve manuscript clarity, but critical issues persist. Firstly, the semi-skeletonised technique remains inadequately defined, described ambiguously as "some arbitrary distance" from LIMA, raising reproducibility concerns. The authors' reply partially addresses this but needs explicit anatomical landmarks for precision. Secondly, statistical interpretation remains problematic. Despite revision, the statistical significance of minimal length differences (0.2 cm) within the standard deviation requires clearer clinical justification, potentially representing type I error. Thirdly, the doubled operative time in the skeletonised group requires deeper analysis beyond superficial acknowledgment; underlying procedural variations or surgeon-specific factors should be clarified further. Lastly, economic analysis remains superficial, relying solely on Ligaclip counts without comprehensive cost breakdown. Recommend: revise thoroughly to address definition precision, statistical robustness, detailed procedural explanations, and comprehensive economic analysis.

Reviewer #3: Dear Authors,

Thank you for your thorough responses and revisions. The manuscript has been significantly improved, and many prior concerns have been addressed. I have only two further suggestions:

1. The term “cost-effectiveness” may overstate the economic implications of your study given the absence of broader economic analyses (e.g., total procedural costs, hospital stay, follow-up costs). Consider revising this term to “procedural efficiency” or “resource utilization efficiency” throughout the manuscript, including the title and abstract.

2. The clinical relevance of early complications (e.g., LIMA spasm, hematoma, re-exploration) is not sufficiently discussed. For instance, you could note the low incidence of these events and the absence of significant between-group differences to support the safety profile of the semi-skeletonized technique.

Thank you again for your efforts.

**Do you want your identity to be public for this peer review?** For information about this choice, including consent withdrawal, please see our Privacy Policy

Reviewer #1: No

Reviewer #2: **Yes: ** Robert J. Chen, MD, MPH

Reviewer #3: **Yes: ** Savvas Lampridis

---

## [Author Response · Author response to Decision Letter 2]

18 May 2025

Response to reviewer's comments

Reviewer #1: I want to thank the authors for their thorough responses to prior comments. I have one comment below followed by some line items.

1. I am still concerned that the central item around which the entire paper revolves is the “semi-skeletonized” LIMA and you are unable to provide a definition that distinguishes it from the only other most commonly used harvest technique which is the pedicled technique. The two most common techniques are skeletonized and pedicled and so it is glaring that this paper does not mention the pedicled technique in any substantial or comparative way. In order to do that, you would have to distinguish the semi-skeletonized and the pedicled and I do not know how you would distinguish those. As you mention, as a cardiac surgeon, they all recognize that minimizing tissue damage during LIMA harvesting is crucial so pedicled LIMAs are usually small (surgeons shy away from taking a large swath of chest wall when taking the pedicle) so how small is small to qualify as semi-skeletonized as opposed to a small pedicle? I think you need to try an attempt at distinguishing the semi-skeletonized vs a pedicled LIMA since even though you do not bring up pedicled LIMAs (other than a few sentences in the Discussion), every surgeon reading this will be wondering what makes your technique different than theirs (if they take small pedicles).

Response: Thanks for your critical appraisal. We have briefly described the semi-skelitonised LIMA technique with appropriate in-text citations (lines 206-212; New Ref number 17). In brief, Horii and Suma have described the semiskeletonised LIMA as a venoarterial pedicle with surrounding thin tissue without muscular support, and keeping the endothoracic fascia attached to the chest wall.

Line 57-60: I am not sure what section this belongs to, but there needs to be a reference noted for all these claims

Response: We understand your concerns. However, as the PLOS One manuscript guidelines don't include those sections, we have deleted the "What is already known on this topic? What this study adds? & How might this study affect research, practice or policy?" sections.

Line 102: to what drainage are you referring?

Response: Apologies for the inconvenience. The sentence has been modified, as highlighted in yellow on line 124.

Line 154-155: this should be mentioned in limitations that you divided (did not randomize) the group into how the LIMA was harvested. It is possible that a significant sorting mechanism was already operating when the patient was chosen for skeletonized vs semi-skeletonized.

Response: Appreciate your comments. There was no sorting mechanism; the LIMA harvesting was conducted randomly. We have modified statements as you suggested, lines 176 & 177, highlighted in yellow.

Line 180: the word “should” should not be here – you are describing what happened, you are not explaining how to do it

Response: Thank you for your attention to detail. We modified the statement as you suggested in line 202 (highlighted in yellow).

Line 280: you are now comparing semi-skeletonized to pedicled – I am not sure of the relevance to this paper as you have not discussed pedicled LIMAs. What is the drainage compared to the skeletonized (the comparison in this paper)?

Response: We understand and appreciate your concerns, so we have removed line 280, as pedicle LIMA is irrelevant to this paper.

Reviewer #2: The authors' revisions significantly improve manuscript clarity, but critical issues persist.

Firstly, the semi-skeletonised technique remains inadequately defined, described ambiguously as "some arbitrary distance" from LIMA, raising reproducibility concerns. The authors' reply partially addresses this but needs explicit anatomical landmarks for precision.

Response: Thanks for your critical appraisal. We have added a brief description of the semi-skelitonised LIMA technique as previously described elsewhere with appropriate in-text citations (lines 206-212; New Ref number 17).

Secondly, statistical interpretation remains problematic. Despite revision, the statistical significance of minimal length differences (0.2 cm) within the standard deviation requires clearer clinical justification, potentially representing type I error.

Response: We defined P <0.05 as significant (not P =0.05), but acknowledge your concerns about P =0.05 being borderline significant, which can occur with a small mean ±SD difference due to the large sample size, and low variability in LIMA length as mean (SD) decreases with increasing sample size and low variability in data, making even minor differences more detectable. However, we acknowledged and discussed these issues in the limitations section with appropriate in-text citations (lines 332-335).

Thirdly, the doubled operative time in the skeletonised group requires deeper analysis beyond superficial acknowledgment; underlying procedural variations or surgeon-specific factors should be clarified further.

Response: We understand your concerns. However, there were no procedural variations as all cases were performed using off-pump CABG techniques and no surgeon-specific factors influenced LIMA harvesting time as all surgeons or trainees harvested both skeletonised and semi-skeletonised LIMA, which we mentioned on page 8, line 233-236.

Lastly, economic analysis remains superficial, relying solely on Ligaclip counts without comprehensive cost breakdown. Recommend: revise thoroughly to address definition precision, statistical robustness, detailed procedural explanations, and comprehensive economic analysis.

Response: Thanks for your suggestions; however, as I said earlier, comprehensive economic analysis was beyond the study objectives, as the current study aimed to compare the 2 LIMA harvest techniques in OPCABG surgery. Further, we modified the “cost-effectiveness” term and replaced it with “resource utilisation efficiency” throughout the manuscript as per the suggestions of Reviewer 3 (yellow highlighted). Nonetheless, we have added a recommendation to conduct further research on the cost-effectiveness of two harvesting techniques, considering factors such as total procedural costs, length of hospital stay, and follow-up expenses (lines 326-329; highlighted in yellow).

Reviewer #3: Dear Authors, Thank you for your thorough responses and revisions. The manuscript has been significantly improved, and many prior concerns have been addressed. I have only two further suggestions:

1. The term “cost-effectiveness” may overstate the economic implications of your study given the absence of broader economic analyses (e.g., total procedural costs, hospital stay, follow-up costs). Consider revising this term to “procedural efficiency” or “resource utilization efficiency” throughout the manuscript, including the title and abstract.

Response: Thanks for your suggestions; we modified the “cost-effectiveness” term and replaced it with “resource utilisation efficiency” throughout the manuscript as per your suggestions (yellow highlighted). Nonetheless, we recommend conducting a further study to evaluate the cost-effectiveness of two harvesting techniques, taking into account the details of total procedural costs, length of hospital stay, and follow-up expenses, etc., as you suggested (lines 326-329; yellow highlighted).

2. The clinical relevance of early complications (e.g., LIMA spasm, hematoma, re-exploration) is not sufficiently discussed. For instance, you could note the low incidence of these events and the absence of significant between-group differences to support the safety profile of the semi-skeletonized technique. Thank you again for your efforts.

Response: Thanks for your suggestions, we have added a brief discussion on these issues on page 11, lines 306-309 (yellow highlighted).

---

## [Decision Letter · Decision Letter 2]

2 Jun 2025

Dear Dr. Ranjan,

Thank you for submitting your manuscript to PLOS ONE. After careful consideration, we feel that it has merit but does not fully meet PLOS ONE’s publication criteria as it currently stands. Therefore, we invite you to submit a revised version of the manuscript that addresses the points raised during the review process.

We look forward to receiving your revised manuscript.

Kind regards,

Usama Waqar, M.B.B.S

Academic Editor

PLOS ONE

Journal Requirements:

Reviewers' comments:

Reviewer's Responses to Questions

**Comments to the Author**

Reviewer #1: All comments have been addressed

Reviewer #2: (No Response)

Reviewer #3: All comments have been addressed

2. Is the manuscript technically sound, and do the data support the conclusions?

Reviewer #1: Yes

Reviewer #2: Partly

Reviewer #3: Yes

3. Has the statistical analysis been performed appropriately and rigorously?

Reviewer #1: I Don't Know

Reviewer #2: Yes

Reviewer #3: Yes

4. Have the authors made all data underlying the findings in their manuscript fully available?

Reviewer #1: Yes

Reviewer #2: Yes

Reviewer #3: Yes

5. Is the manuscript presented in an intelligible fashion and written in standard English?

Reviewer #1: Yes

Reviewer #2: Yes

Reviewer #3: Yes

Reviewer #1: PLOS ONE

Safety and cost-effectiveness of semi-skeletonised versus skeletonised left internal mammary artery harvesting techniques: the BANGABANDHU study

PONE-D-25-07678R2

May 30, 2025

I want to thank the authors for their responses.

Overall, I have a couple of items left.

1. The last sentence of the discussion says “cost-effectiveness” and this word should be changed as this study is not a cost-effectiveness analysis

2. Unless skeletonized vs semi-skeletonized were chosen by a randomizing mechanism, the claim that the use of either technique was randomized is not accurate and should be removed. Without a randomizing mechanism, the choice is up to the surgeon and they choose which intervention to give and that is not accepted as being randomized.

Reviewer #2: Despite these improvements in the revision, several critical issues remain unresolved:

The “semi-skeletonised” LIMA technique is still vaguely defined. The revision cites Horii’s description of a thin venoarterial pedicle but omits precise anatomical boundaries. Without clear landmarks, the method is clearly indistinguishable from a minimal pedicled harvest, undermining reproducibility.

A 0.2 cm LIMA length difference (p≈0.05) is statistically marginal and clinically negligible. Treating this borderline result as significant overstates its importance and likely reflects sample-size effects.

The skeletonised harvest time was double the semi-skeletonised time, yet this is unexplained. Claiming uniform technique and staff ignores potential learning-curve or procedural complexity factors behind such a large efficiency gap.

No substantive economic analysis is presented. Merely counting Ligaclips and renaming outcomes as “resource utilization” falls short of a genuine cost comparison, contrary to the manuscript’s framing.

Reviewer #3: Dear Authors,

Thank you for considering my suggested revisions. Overall, you have put a lot of effort into revising your work by integrating the feedback provided. I believe the resulting changes have significantly improved the rigor and overall quality of your manuscript.

Congratulations on your work.

**Do you want your identity to be public for this peer review?** For information about this choice, including consent withdrawal, please see our Privacy Policy

Reviewer #1: No

Reviewer #2: **Yes: ** Robert J. Chen, MD, MPH

Reviewer #3: **Yes: ** Savvas Lampridis

---

## [Author Response · Author response to Decision Letter 3]

19 Jun 2025

Dear Editor-in-Chief

PLOS ONE

Thank you for considering our manuscript for publication in PLOS ONE. I am attaching our response to the reviewer comments and point-by-point clarifications for your review.

Re: Response to Review Comments on PONE-D-25-07678R2

Reviewer #1: I want to thank the authors for their responses. Overall, I have a couple of items left.

1. The last sentence of the discussion says “cost-effectiveness” and this word should be changed as this study is not a cost-effectiveness analysis.

Response: Thank you very much for your attention to detail. The statement has been modified as you suggested, with the changes highlighted in yellow (line 337).

2. Unless skeletonized vs semi-skeletonized were chosen by a randomizing mechanism, the claim that the use of either technique was randomized is not accurate and should be removed. Without a randomizing mechanism, the choice is up to the surgeon and they choose which intervention to give and that is not accepted as being randomized.

Response: We appreciate your constructive comments. Although we randomly performed either skeletonised or semi-skeletonised techniques, we agreed that the sampling procedures didn't comply with the full structure of a randomised trial, which is why we modified the sentences as you suggested (highlighted in yellow, line 177).

Reviewer #2: Despite these improvements in the revision, several critical issues remain unresolved:

1. The “semi-skeletonised” LIMA technique is still vaguely defined. The revision cites Horii’s description of a thin venoarterial pedicle but omits precise anatomical boundaries. Without clear landmarks, the method is clearly indistinguishable from a minimal pedicled harvest, undermining reproducibility.

Response: Dear Reviewer, to the best of our knowledge, Horii's description of semi-skeletonised LIMA is precise and clear, and we have precisely defined the anatomical boundaries in lines 204-212 & 215-218 (yellow highlighted). Unfortunately, neither do we have an idea nor have we found literature defining "minimal pedicled harvest" techniques.

2. A 0.2 cm LIMA length difference (p≈0.05) is statistically marginal and clinically negligible. Treating this borderline result as significant overstates its importance and likely reflects sample-size effects.

Response: We agree that a p-value of 0.05 is not statistically significant, as noted in the methodology section. A p-value of <0.05 was considered statistically significant (highlighted in the yellow, lines 244 & 245). However, we utilise independent variables with a p-value of ≤0.05 in the multivariate LR model to avoid missing potentially important effects, e.g. false negative. Nevertheless, we have already acknowledged your concerns as a study limitation in lines 332-335, highlighted in yellow with appropriate citations.

3. The skeletonised harvest time was double the semi-skeletonised time, yet this is unexplained. Claiming uniform technique and staff ignores potential learning-curve or procedural complexity factors behind such a large efficiency gap.

Response: We understand your concerns, which we have already explained in the discussion section (Yellow highlighted, lines 282-285). Furthermore, the difference in harvesting time may be associated with the learning curve or procedural complexity; however, we confirmed that all surgeons were equally qualified and performed both harvesting techniques (Yellow highlighted, lines 233-236). Despite the risk of bias, a large sample like ours minimises the risk of bias and confirms the generalisability of the findings.

4. No substantive economic analysis is presented. Merely counting Ligaclips and renaming outcomes as “resource utilization” falls short of a genuine cost comparison, contrary to the manuscript’s framing.

Response: We appreciate your concerns; however, in LMICs like Bangladesh, saving ~25 ligaclip per case indicates significant savings at the end of the year. For example, 25 Ligaclips per case saves approximately $100, which means that, annually in Bangladesh, we perform around 8,000 CABG surgeries, resulting in a potential annual savings of approximately $800,000.00. This is a significant amount for both the Bangladeshi people and the government. I understand your work in the NHS, which is well-organised and fully supported by the UK government; however, a simple modification of the skeletonised technique might help people in LMICs significantly.

However, as you raised concerns and as 3rd reviewer suggested, "cost-effectiveness" terms were replaced with "resource utilisation", and in the absence of a comprehensive economic analysis, we recommended further research on the cost-effectiveness of two harvesting techniques, considering factors such as total procedural costs, length of hospital stay, and follow-up expenses (lines 326-329; highlighted in yellow).

Reviewer #3: Dear Authors,

Thank you for considering my suggested revisions. Overall, you have put a lot of effort into revising your work by integrating the feedback provided. I believe the resulting changes have significantly improved the rigor and overall quality of your manuscript.

Congratulations on your work.

Response: Thank you very much for your constructive comments and valuable suggestions, which greatly enhance the manuscript's quality.

We hope this revised version meets your standards for publication approval.

With thanks and regards

Dr Redoy Ranjan

Associate Editor, PLOS ONE

---

## [Decision Letter · Decision Letter 3]

14 Jul 2025

Dear Dr. Ranjan,

Thank you for submitting your manuscript to PLOS ONE. After careful consideration, we feel that it has merit but does not fully meet PLOS ONE’s publication criteria as it currently stands. Therefore, we invite you to submit a revised version of the manuscript that addresses the points raised during the review process.

We look forward to receiving your revised manuscript.

Kind regards,

Eyüp Serhat Çalık

Academic Editor

PLOS ONE

Journal Requirements:

Additional Editor Comments:

I am grateful to the distinguished authors for their revisions done and appropriate responses. Below are some additional suggestions for the manuscript. Please resubmit your manuscript with your point-by-point answers and corrections as soon as possible.

Reviewers' comments:

Reviewer's Responses to Questions

**Comments to the Author**

Reviewer #1: All comments have been addressed

Reviewer #2: All comments have been addressed

Reviewer #3: All comments have been addressed

2. Is the manuscript technically sound, and do the data support the conclusions?

Reviewer #1: Partly

Reviewer #2: Yes

Reviewer #3: (No Response)

3. Has the statistical analysis been performed appropriately and rigorously?

Reviewer #1: I Don't Know

Reviewer #2: Yes

Reviewer #3: (No Response)

4. Have the authors made all data underlying the findings in their manuscript fully available?

Reviewer #1: No

Reviewer #2: Yes

Reviewer #3: (No Response)

5. Is the manuscript presented in an intelligible fashion and written in standard English?

Reviewer #1: Yes

Reviewer #2: Yes

Reviewer #3: (No Response)

Reviewer #1: PLOS ONE

Safety and cost-effectiveness of semi-skeletonised versus skeletonised left internal mammary artery harvesting techniques: the BANGABANDHU study

PONE-D-25-07678R3

July 2, 2025

I want to thank the authors for answering my questions. I believe that the paper is much better and helps correctly place the findings as the outcomes of a single institutional experience. I have a couple general comments below followed by some line items.

First, it should be acknowledged that your institutional preference is use Ligaclips, but many other surgeons use clips on reusable clip appliers and when using this latter method, the cost difference is likely negligible.

Second, were these all primary operations or were some redo’s? This would provide a major source of bias and so it should be addressed.

Line 205-209: it is a little unclear here with the first sentence stating “semi-skeletonised LIMA technique preserved……” followed by the subsequent sentence stating “semi-skeletonized LIMA was excised as…..” which included structures that it said were preserved in the prior sentence. These two sentences can be written more clearly to let the reader know what was the exact technique.

Line 219: “we tried to avoid opening the pleural space”

Line 231: “in both cases, we opened the left pleura to enhance LIMA visualization”

These are contradictory statements.

Line 237: you have to remove the word “randomly” here – it was not randomized. You changed it in Line 177 and it needs to be changed here too.

Line 286: “However” seems to refer to the prior sentence (semi-skeletonized) and the “However” sentence seems to refer to the skeletonized technique so the two sentences are combined together with “however” incorrectly.

Line 308: What was your definition of hematoma? Leaking of vasovasorum over a certain distance? Impeding blood flow? And what was your definition of spasm?

Line 330: I would leave out “favoring semi-skeletonized techniques” – you recommend doing future studies and you do not know the outcomes (hence the need for the studies) so you should not say that they are going to favor one technique or the other.

Reviewer #2: The authors have diligently addressed the previous reviewers' concerns.

The manuscript has been revised to replace "cost-effectiveness" with the more accurate "resource utilisation efficiency," and the inaccurate claim of randomization was corrected to "stochastically performed." The authors clarified the surgical technique by providing more precise anatomical descriptions and procedural details. They also appropriately acknowledged the marginal statistical significance of the LIMA length difference as a study limitation.

The statistical analysis is sound, and the English is suitable for publication. The authors provided compelling arguments for their findings, particularly regarding the significant differences in harvesting time and clip usage between the two methods.

Reviewer #3: Dear Authors,

Although I did not provide specific suggestions in the previous round, I appreciate the revisions you made in response to the comments from the other reviewers. These changes have further strengthened the manuscript and clearly positioned it as a resource utilization analysis.

I wish you every success with the publication of your work.

**Do you want your identity to be public for this peer review?** For information about this choice, including consent withdrawal, please see our Privacy Policy

Reviewer #1: No

Reviewer #2: **Yes: ** Robert J. Chen, MD, MPH

Reviewer #3: **Yes: ** Savvas Lampridis

---

## [Author Response · Author response to Decision Letter 4]

17 Jul 2025

Dear Editor-in-Chief

PLOS ONE

Thank you for considering our manuscript for publication in PLOS ONE. I am attaching our response to the reviewer comments and point-by-point clarifications for your review.

Re: Response to Review Comments on PONE-D-25-07678R3

Reviewer #1: I want to thank the authors for answering my questions. I believe that the paper is much better and helps correctly place the findings as the outcomes of a single institutional experience. I have a couple general comments below followed by some line items.

1. First, it should be acknowledged that your institutional preference is use Ligaclips, but many other surgeons use clips on reusable clip appliers and when using this latter method, the cost difference is likely negligible.

Response: We appreciate your concerns; however, in Bangladesh, all surgeons and institutes utilise Ligaclips with reusable clip applicators, which I mentioned in line 252. Further, I believe you overlooked my explanation in the previous version, as the same concerns were raised by Reviewer 2.

Although "the cost difference is likely negligible", per case, like you said, saving ~25 ligaclip per case indicates significant savings at the end of the year. For example, 25 Ligaclips per case saves approximately $100, which means that, annually in Bangladesh, we perform around 8,000 CABG surgeries, resulting in a potential annual savings of roughly $800,000.00. This is a significant amount for both the Bangladeshi people and the government. I appreciate your work at a well-organised institute supported by the government; however, a simple modification of the skeletonised LIMA technique might significantly help people in LMICs.

2. Second, were these all primary operations or were some redo’s? This would provide a major source of bias and so it should be addressed.

Response: This study included isolated primary (elective and emergency) CABG procedures, including redo CABG cases if the LIMA is available for harvesting. I have revised the methodology section as you recommended for better clarity on pages 6 and 8, specifically lines 175-176 and lines 249-251. For both primary and redo CABG, the harvesting techniques for the LIMA remain consistent, as outlined in the LIMA harvesting technique section on pages 7-9. This consistency helps mitigate the risk of bias. Additionally, we have added a limitations statement regarding this concern on page 13, lines 361-364.

3. Line 205-209: it is a little unclear here with the first sentence stating “semi-skeletonised LIMA technique preserved……” followed by the subsequent sentence stating “semi-skeletonized LIMA was excised as…..” which included structures that it said were preserved in the prior sentence. These two sentences can be written more clearly to let the reader know what the exact technique was.

Response: We apologise for the inconvenience; we have revised the statement for improved clarity and presentation in lines 219-222.

4. Line 219: “we tried to avoid opening the pleural space”. Line 231: “in both cases, we opened the left pleura to enhance LIMA visualization”. These are contradictory statements.

Response: Thank you for your careful attention to detail. To improve clarity, we have removed line 219, which stated, "we tried to avoid opening the pleural space." We have retained line 231 (now line 243): "In both cases, we opened the left pleura to enhance LIMA visualisation."

5. Line 237: you have to remove the word “randomly” here – it was not randomized. You changed it in Line 177 and it needs to be changed here too.

Response: We have now replaced "randomly" with "stochastically" on line 249; the change is highlighted in yellow.

6. Line 286: “However” seems to refer to the prior sentence (semi-skeletonized) and the “However” sentence seems to refer to the skeletonized technique so the two sentences are combined together with “however” incorrectly.

Response: Thank you for your attention to detail. The statement has been modified for improved clarity and presentation, as indicated in line 310 (highlighted in yellow).

7. Line 308: What was your definition of hematoma? Leaking of vasovasorum over a certain distance? Impeding blood flow? And what was your definition of spasm?

Response: There is currently no standard distance threshold that defines a LIMA hematoma or spasm. I have briefly described the LIMA hematoma and spasm in lines 196-209. Additionally, I included the current study strategies for managing cases of LIMA hematoma or spasm and outlined our approach to dealing with these issues.

8. Line 330: I would leave out “favoring semi-skeletonized techniques” – you recommend doing future studies and you do not know the outcomes (hence the need for the studies) so you should not say that they are going to favor one technique or the other.

Response: I appreciate your attention to detail. As you suggested, I have removed "favouring semi-skeletonised techniques" from lines 355.

Reviewer #2: The authors have diligently addressed the previous reviewers' concerns.

The manuscript has been revised to replace "cost-effectiveness" with the more accurate "resource utilisation efficiency," and the inaccurate claim of randomization was corrected to "stochastically performed." The authors clarified the surgical technique by providing more precise anatomical descriptions and procedural details. They also appropriately acknowledged the marginal statistical significance of the LIMA length difference as a study limitation. The statistical analysis is sound, and the English is suitable for publication. The authors provided compelling arguments for their findings, particularly regarding the significant differences in harvesting time and clip usage between the two methods.

Response: We sincerely thank the reviewer for their kind words and thoughtful evaluation. We appreciate the recognition of the attention to detail in our work, and we are grateful for the positive feedback and encouragement. It reinforces our commitment to maintaining high standards in both methodology and presentation.

Reviewer #3: Dear Authors,

Although I did not provide specific suggestions in the previous round, I appreciate the revisions you made in response to the comments from the other reviewers. These changes have further strengthened the manuscript and clearly positioned it as a resource utilization analysis.

I wish you every success with the publication of your work.

Response: We sincerely thank the reviewer for their thoughtful and encouraging comments. We greatly appreciate your recognition of the strengths of our work and your emphasis on the importance of attention to detail. Your positive feedback reinforces our commitment to scientific rigour, and we have carefully reviewed the manuscript to ensure consistency, clarity, and precision throughout.

We hope this revised version meets your standards for publication approval.

With thanks and regards

Dr Redoy Ranjan

Associate Editor, PLOS ONE

Institute of Cardiovascular Research, Royal Holloway University of London, London, UK

Department of Cardiac Surgery, St George’s University Hospitals NHS Foundation Trust, UK

Department of Cardiac Surgery, Bangabandhu Sheikh Mujib Medical University, Bangladesh Email: redoy_ranjan@bsmmu.edu.bd

---

## [Decision Letter · Decision Letter 4]

14 Aug 2025

Safety and resource utilisation efficiency of semi-skeletonised versus skeletonised left internal mammary artery harvesting techniques: the BANGABANDHU study

PONE-D-25-07678R4

Dear Dr. Ranjan,

We’re pleased to inform you that your manuscript has been judged scientifically suitable for publication and will be formally accepted for publication once it meets all outstanding technical requirements.

Kind regards,

Eyüp Serhat Çalık

Academic Editor

PLOS ONE

Additional Editor Comments (optional):

Dear Authors, I am pleased to inform you that your manuscripts are now ready for acceptance. Your paper has been re-evaluated by reviewer 1, and I kindly request that you make a few minor grammatical corrections as suggested by the reviewer after acceptance. Best wishes for your success.

Reviewers' comments:

Reviewer's Responses to Questions

**Comments to the Author**

Reviewer #1: All comments have been addressed

2. Is the manuscript technically sound, and do the data support the conclusions?

Reviewer #1: Yes

3. Has the statistical analysis been performed appropriately and rigorously?

Reviewer #1: I Don't Know

4. Have the authors made all data underlying the findings in their manuscript fully available?

Reviewer #1: No

5. Is the manuscript presented in an intelligible fashion and written in standard English?

Reviewer #1: Yes

Reviewer #1: Line 110: grammar needs to be corrected (“which is commonly utilized” referring to IMA)

Line 119: need to qualify Ligaclip that it is in your institution when using Ligaclip (many/maybe majority of programs do not use a Ligaclip when taking down the IMA) – you state this as if it is a general rule that skeletonized IMA harvest uses more Ligaclip and that is not true since likely most places do not use the Ligaclip for this harvest. You also say your Ligacliip is reusable. You should put in parentheses which device you are using and who makes it.

Line 128: why is this “paradoxical?”

Line 252: the paper is in the past tense and you use the present tense here

Line 331: RCT is not a proper name and does not need to be capitalized

**Do you want your identity to be public for this peer review?** For information about this choice, including consent withdrawal, please see our Privacy Policy

Reviewer #1: No

---

## [Editor Report · Acceptance letter]

PONE-D-25-07678R4

PLOS ONE

Dear Dr. Ranjan,

I'm pleased to inform you that your manuscript has been deemed suitable for publication in PLOS ONE. Congratulations! Your manuscript is now being handed over to our production team.

Kind regards,

on behalf of

Dr. Eyüp Serhat Çalık

Academic Editor

PLOS ONE